# Sites of vulnerability on ricin B chain revealed through epitope mapping of toxin-neutralizing monoclonal antibodies

**David J. Vance[1]\*, Amanda Y. Poon[2], Nicholas J. Mantis[1,2]\***

**1** Division of Infectious Disease, New York State Department of Health, Wadsworth Center, Albany, NY, United States of America, **2** Department of Biomedical Sciences, University at Albany School of Public Health, Albany, NY, United States of America

\* david.vance@health.ny.gov (DJV); nicholas.mantis@health.ny.gov (NJM)

**Data Availability Statement:** All relevant data are within the paper and its Supporting Information files.

## Abstract

Ricin toxin's B subunit (RTB) is a multifunctional galactose (Gal)-/N-acetylgalactosamine (GalNac)-specific lectin that promotes uptake and intracellular trafficking of ricin's ribosome-inactivating subunit (RTA) into mammalian cells. Structurally, RTB consists of two globular domains (RTB-D1, RTB-D2), each divided into three homologous sub-domains (α, β, γ). The two carbohydrate recognition domains (CRDs) are situated on opposite sides of RTB (sub-domains 1α and 2γ) and function non-cooperatively. Previous studies have revealed two distinct classes of toxin-neutralizing, anti-RTB monoclonal antibodies (mAbs). Type I mAbs, exemplified by SylH3, inhibit (~90%) toxin attachment to cell surfaces, while type II mAbs, epitomized by 24B11, interfere with intracellular toxin transport between the plasma membrane and the trans-Golgi network (TGN). Localizing the epitopes recognized by these two classes of mAbs has proven difficult, in part because of RTB's duplicative structure. To circumvent this problem, RTB-D1 and RTB-D2 were expressed as pIII fusion proteins on the surface of filamentous phage M13 and subsequently used as "bait" in mAb capture assays. We found that SylH3 captured RTB-D1 (but not RTB-D2) in a dose-dependent manner, while 24B11 captured RTB-D2 (but not RTB-D1) in a dose-dependent manner. We confirmed these domain assignments by competition studies with an additional 8 RTB-specific mAbs along with a dozen a single chain antibodies ($V_H$Hs). Collectively, these results demonstrate that type I and type II mAbs segregate on the basis of domain specificity and suggest that RTB's two domains may contribute to distinct steps in the intoxication pathway.

## Introduction

The plant toxin, ricin, is classified by military and public health officials as a biothreat agent because of its extreme potency following inhalation, coupled with the ease by which the toxin can be procured from castor beans (*Ricinus communis*) [1]. Ricin's A and B subunits each contribute to toxicity. The A subunit (RTA) is a ribosome-inactivating protein (RIP) that functions by depurination of a conserved adenine residue within the sarcin-ricin loop (SRL) of 28S rRNA [2, 3]. The B subunit (RTB) is a galactose (Gal)- and N-acetylgalactosamine (GalNAc)-

**Funding:** This work was supported by NIAID contract No. HHSN272201400021C and grant AI125190 to NJM. The content is solely the responsibility of the authors and does not necessarily represent the official views of the National Institutes of Health. The funders had no role in study design, data collection and analysis, decision to publish, or preparation of the manuscript.

**Competing interests:** The authors have declared that no competing interests exist.

specific lectin capable of binding to surface exposed glycoproteins and glycolipids, including on the surface of cells in the lung [4]. Following endocytosis, RTB mediates retrograde transport of ricin to the trans-Golgi network (TGN) and the endoplasmic reticulum (ER). Within the ER, RTA is liberated from RTB and retrotranslocated across the ER membrane into the cytoplasm, where it refolds and interacts with its substrate with remarkable efficiency [5–8].

Structurally, RTB consists of two globular domains with identical folding topologies (**Fig 1**) [9, 10]. The two domains, RTB-D1 and RTB-D2, are each further apportioned into three homologous sub-domains (α, β, γ) that likely arose as a result of gene duplication of a primordial carbohydrate recognition domain (CRD) [11]. X-ray crystallography (PDB ID 2AAI) [11], site-directed mutagenesis [12–14] and phage display of RTB-D1 and RTB-D2 [15] has revealed that each domain retains functional carbohydrate recognition activity. Specifically, sub-domain 1α binds Gal and is considered the "low affinity" CRD, while sub-domain 2γ binds Gal and GalNac and is considered a "high affinity" CRD [12, 16–18]. Both domains contribute to cell attachment and toxin uptake [12–14].

In light of its essential role in toxin uptake and trafficking, RTB is an obvious target to consider in the development of ricin countermeasures [19]. Indeed, in reports dating back more than a decade, we and others have described collections of RTB-specific monoclonal antibodies (mAbs) that have been evaluated for toxin-neutralizing activities in cell-based assays and, in some cases, mouse models of ricin intoxication [20–27]. Overall, relatively few RTB-specific mAbs capable of passively protecting mice against systemic or intranasal ricin challenge have been described. However, in our collection, two stand out: SylH3 and 24B11. SylH3 is classified as a type I mAb because it is highly effective at blocking ricin attachment to cell surfaces, suggesting it neutralizes ricin by preventing ricin uptake [26–28]. 24B11, on the other hand, has little impact on attachment. Rather, it appears to neutralize ricin by interfering with intracellular trafficking between the plasma membrane and the TGN. We have classified 24B11 as a type II mAb [25].

Competition studies have demonstrated that 24B11 and SylH3 recognize different epitopes on RTB, although the location of those epitopes remains to be determined. 24B11's epitope was tentatively assigned to RTB-D1, based on limited reactivity with affinity-enriched phage displayed peptides [21]. SylH3's epitope was tentatively assigned to RTB-D2 by virtue of the fact that RTB's high affinity CRD is situated within subdomain 2γ [26–28]. The duplicative nature of RTB, structurally and functionally, has made epitope localization studies challenging. Further complicating epitope mapping studies have been challenges associated with expression of soluble, recombinant RTB in *E.coli* [29, 30]. However, full-length RTB (FL-RTB), as well RTB-D1 and RTB-D2 constructs, have been successfully expressed as fusion proteins on the tip of filamentous phage M13 [15]. With our past expertise in phage display, we reasoned that RTB domain display might offer a highly effective means by which to localize epitopes recognized by 24B11 and SylH3, as well as other RTB-specific mAbs in our collection.

## Methods

### Chemicals and biological reagents

Labeled and unlabeled ricin toxin (*Ricinus communis* agglutinin II;RCA-II) was purchased from Vector Laboratories (Burlingame, CA, USA). Unless noted otherwise, all of the other chemicals were obtained from Sigma-Aldrich, Inc. (St. Louis, MO, USA).

### Ricin-specific mAbs and $V_H$Hs

The murine mAbs were purified from hybridoma supernatants by Protein A chromatography at the Dana Farber Cancer Institute monoclonal antibody core facility (Boston, MA, USA) [24,

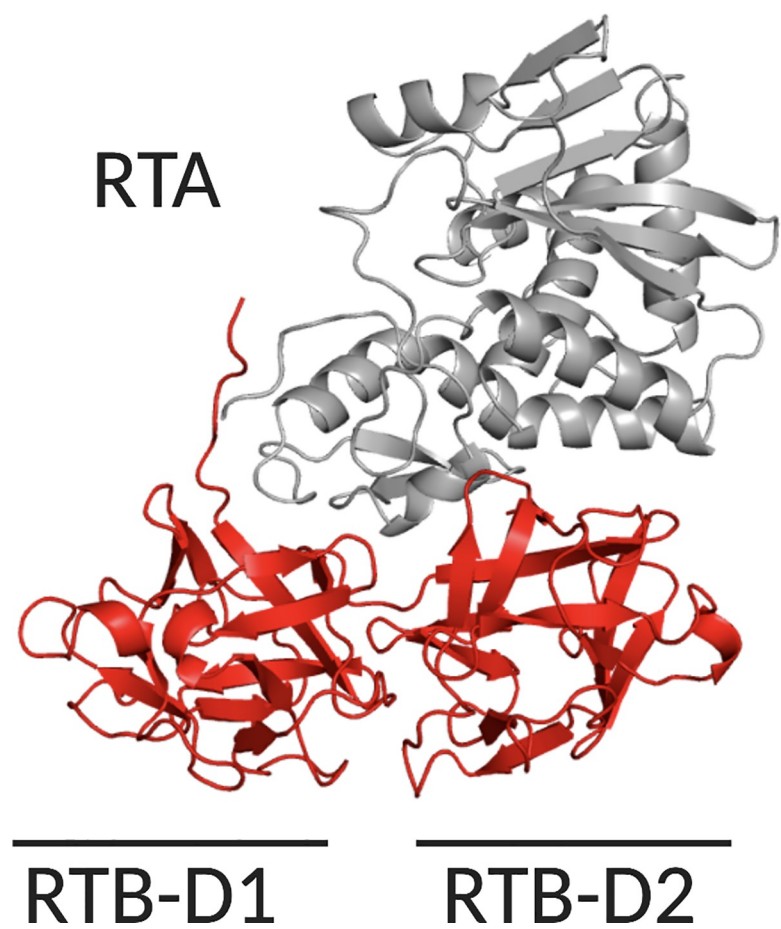

**Fig 1. Structure of ricin's enzymatic (RTA) and binding (RTB) subunits.** Cartoon representation of ricin holotoxin (PDB ID 2AAI) with RTA (gray) and RTB's (red) two domains (RTB-D1, RTB-D2) highlighted.

31]. The ricin-specific, single-domain antibodies ($V_HH$) were purified as described [32, 33]. The $V_H$Hs used in this study carry a C-terminus E epitope tag (E-tag; GAPVPYPDPLEPR) for the purpose of detection by ELISA using HRP-conjugated, affinity-purified anti-E-tag goat IgG.

## Phage display of RTB domains 1 (RTB-D1) and 2 (RTB-D2)

A pET-15b plasmid encoding RTB cDNA (pRTB) was provided by Dr. Paul Sehnke (University of Florida). Primers were designed to amplify either full length RTB (RTB-FL) or individual domains (RTB-D1, RTB-D2) (**S1 Table**). We defined RTB-D1 as residues 1–135 and RTB-D2 as 136–262 (**S2 Table**) [11, 17]. Forward primers were designed to include a 5' NotI site, while reverse primers contained a 5' AscI site. The codon encoding cysteine at RTB position 4 (5'-TGT-3'), normally involved in disulfide bond formation with RTA, was changed to serine (5'-AGT-3') to avoid unwanted oxidation and misfolding in the RTB phage products. The RTB amplicons were cloned into the JSC phagemid (GenBank EU109715;[32, 34]) using sticky-end ligation (*Not*I, *Asc*I) to encode N-terminal pIII fusion proteins. The resulting plasmids were transformed into NEB® Turbo Competent E. coli (New England Biolabs, Ipswich, MA). The Turbo Competent strain contains an amber suppressor gene for proper translation of the RTB-pIII products, as well as the F' plasmid necessary for helper phage superinfection.

To produce M13 phage bearing RTB or its domains, *E.coli* from each transformation were infected with VCSM13 helper phage (kindly provided by Chuck Shoemaker, Tufts University). Stationary phase cultures were then subjected to centrifugation and the supernatants were treated with 20% PEG8000/2.5M NaCl to precipitate M13 phage. Resulting phage pellets were reconstituted in PBS and titered on *E. coli* ER2738 (New England Biolabs).

## ELISAs

The competitive ELISA protocol known as EPICC has been described [35]. Nunc Maxisorb F96 microtiter plates (ThermoFisher Scientific, Pittsburgh, PA, USA) were coated overnight with capture mAb (1 μg/mL) in PBS [pH 7.4]. Plates were blocked with 2% goat serum, washed, and incubated with biotinylated-ricin, in the absence or presence of analyte mAbs (10 μg/mL). The amount of biotinylated-ricin was adjusted to achieve the $EC_{90}$ of each capture antibody. After 1 h, the plates were washed and developed with streptavidin-HRP antibody (1:1000; SouthernBiotech, Birmingham, AL, USA) and 3,3′,5,5′-tetramethylbenzidine (TMB; Kirkegaard & Perry Labs, Gaithersburg, MD, USA). The plates were analyzed with a Versamax spectrophotometer equipped with Softmax Pro 7 software (Molecular Devices, Sunnyvale, CA, USA).

For $V_H H$ competition assays, Nunc Maxisorb F96 microtiter plates were coated overnight with the capture mAbs (1 μg/mL). Plates were blocked with 2% goat serum, washed, and incubated with ricin (1 μg/mL) for 1 h. The plates were washed, overlaid with $V_H Hs$ (330 nM; ~10 μg/mL) for 1 h, then washed again and probed with anti-E-tag-HRP secondary antibody (1:10000; Bethyl Labs, Montgomery, TX). The plates were developed with TMB, as described above for the ELISAs. To estimate the binding of the $V_H Hs$ to the remaining capture mAbs, we arbitrarily set maximal ricin toxin binding (100%) as the highest $OD_{450}$ value observed among the panel of capture mAbs calculated as: % $V_H H$ binding = [(observed $OD_{450}$)/(maximal $OD_{450}$)] x 100.

For M13 phage ELISAs, Nunc Maxisorb F96 microtiter plates were coated overnight with mAb or $V_H H$ (1 μg/mL in PBS). Plates were blocked with 2% bovine serum albumin (BSA) in PBS and then incubated with 5 x $10^{11}$ plaque forming units (PFU) per mL in PBS (5 x $10^{10}$ PFU per well) of each of the three phage constructs. The plates were washed to remove unbound phages and then probed with anti-M13-HRP IgG secondary antibody (1:5000, Cytiva [formerly GE Healthcare], Marlboro, MA) followed by TMB as noted above.

For RTB-D1/D2 competition ELISAs, Nunc Maxisorb F96 microtiter plates were coated overnight with indicated mAb (1 μg/mL in PBS) then blocked with 2% (w/v) BSA in PBS. Wells were treated with 10 μg/mL of ricin, RTA or RTB for 1 h, washed and then probed with 5 x $10^{10}$ PFU per well of RTB-D1 or RTB-D2. The plates were washed to remove unbound phages and developed with anti-M13-HRP IgG (1:5000) and TMB as noted above.

## Statistical analyses

Statistical analyses were performed in GraphPad Prism version 8 (GraphPad Software, San Diego, CA, USA).

## Modeling of ricin toxin

Images of ricin holotoxin (PBD ID 2AAI) were generated using PyMOL (The PyMOL Molecular Graphics System, Schrodinger LLC, San Diego, CA, USA).

## Results

### Competition ELISA reveal distinct epitope clusters on RTB

The relative epitope locations of the 10 RTB-specific toxin-neutralizing (underlined throughout) and non-neutralizing mAbs in our collection are not known, because the collection as a whole has never been subject to cross-competition assays. To address this issue, we performed competitive sandwich ELISAs in which capture mAbs were immobilized on microtiter plates and then probed with biotinylated ricin in the presence of molar excess competitor mAb (Table 1; Fig 2A). The ELISAs revealed three competition groups of epitopes that we referred to as clusters 5, 6 and 7. Clusters 1–4 have already been described on RTA [31]. Cluster 5 consisted of SylH3 and JB4, cluster 6 consisted of 24B11, MH3, 8A1, JB11, BJF9, and LF1, and cluster 7 only LC5. 8B3 competed with both SylH3 and 24B11, tentatively situating its epitope between clusters 5 and 6. Thus, the cross-competition ELISA defined at least three spatially distinct epitope clusters on RTB with toxin-neutralizing mAbs.

### Epitope localization of RTB-specific mAbs using phage-displayed RTB domains 1 and 2

While the competition ELISAs enabled us to assign the mAbs to different clusters, the relative locations of those clusters on RTB remains undefined. In previous studies, we situated 24B11's epitope on RTB-D1 and SylH3's epitope on RTB-D2, although those assignments were highly speculative [21, 24, 26, 27]. As a more definitive strategy to localize B cell epitopes on individual domains of RTB, we expressed full-length RTB (RTB-FL; residues 1–262) or the individual RTB domains, RTB-D1 (residues 1–135) and RTB-D2 (residues 136–262), as N-terminal pIII fusion proteins on the surface of phage M13 [15]. We then employed the three M13 phage constructs as "bait" in capture ELISAs with the panel of 10 anti-RTB mAbs (Table 1). WECB2, an RTA-specific mAb, was used as a control for these ELISAs.

**Table 1. Domain Assignments of RTB-specific mAbs.**

| mAb[a] | Cluster | RTB capture | | | Competition | | Domain |
| --- | --- | --- | --- | --- | --- | --- | --- |
| | | FL[b] | D1[b] | D2[b] | SylH3 | 24B11 | Assignment[c] |
| SylH3 | 5 | + | ++++ | - | Y | N | D1 |
| JB4 | 5 | ++ | ++++ | (++++) | Y | N | D1 |
| 24B11 | 6 | ++ | - | ++++ | N | Y | D2 |
| MH3 | 6 | ++ | - | ++++ | N | Y | D2 |
| 8A1 | 6 | ++ | - | ++++ | N | Y | D2 |
| JB11 | 6 | +++ | +++ | ++++ | N | Y | D1 + D2 |
| BJF9 | 6 | ++ | (++) | ++++ | N | Y | D2 |
| LF1 | 6 | + | - | +/- | N | Y | D2 |
| 8B3 | 5/6 | ++ | - | - | Y | Y | D1-D2 |
| LC5 | 7 | ++ | - | ++++ | N | N | D2 |
| WECB2 | 1 | - | - | - | n.d. | n.d. | RTA |

[a], underlines indicate mAbs with toxin-neutralizing activity in the Vero cell cytotoxicity assay

[b], The plus signs (+, ++, +++, etc) summarize the relative amount of RTB capture (RTB-FL; RTB-D1, RTB-D2) from results presented in Fig 2. Parentheses indicate putative subordinate or secondary

[c], "D1-D2" indicates proposed epitope at the D1-D2 interface, while D1 + D2 indicates independent epitopes on each domain. Abbreviations: Y, yes; N, no; n.d., not determined; FL, full length.

**A.**

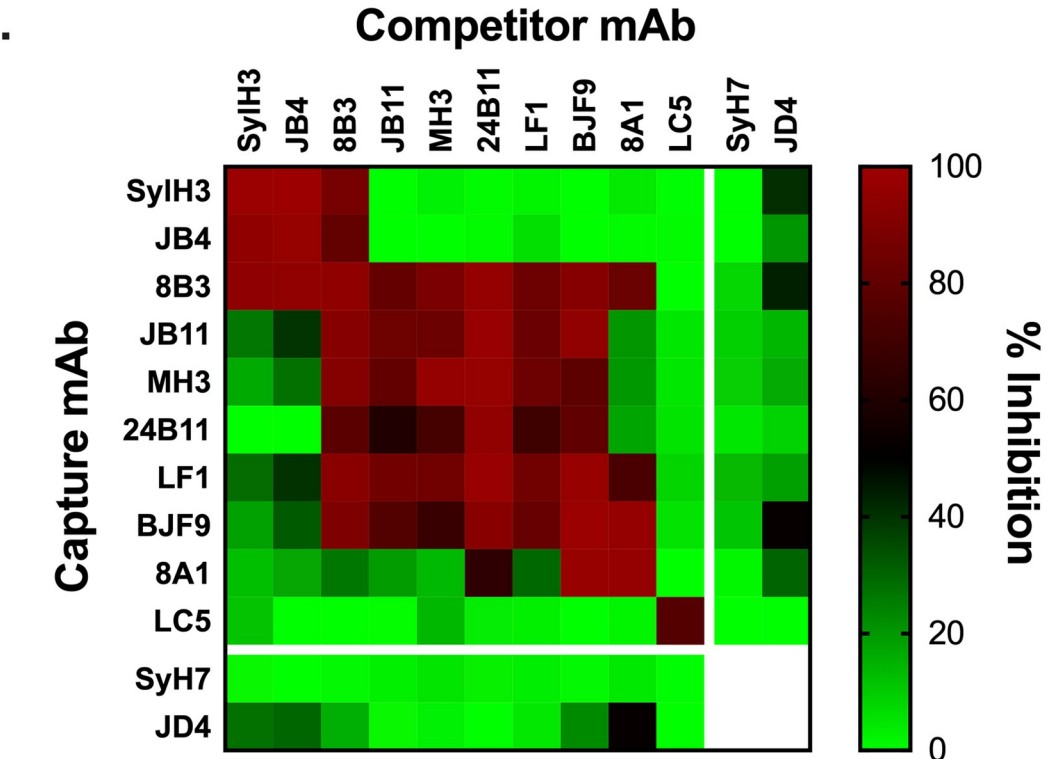

**B.**

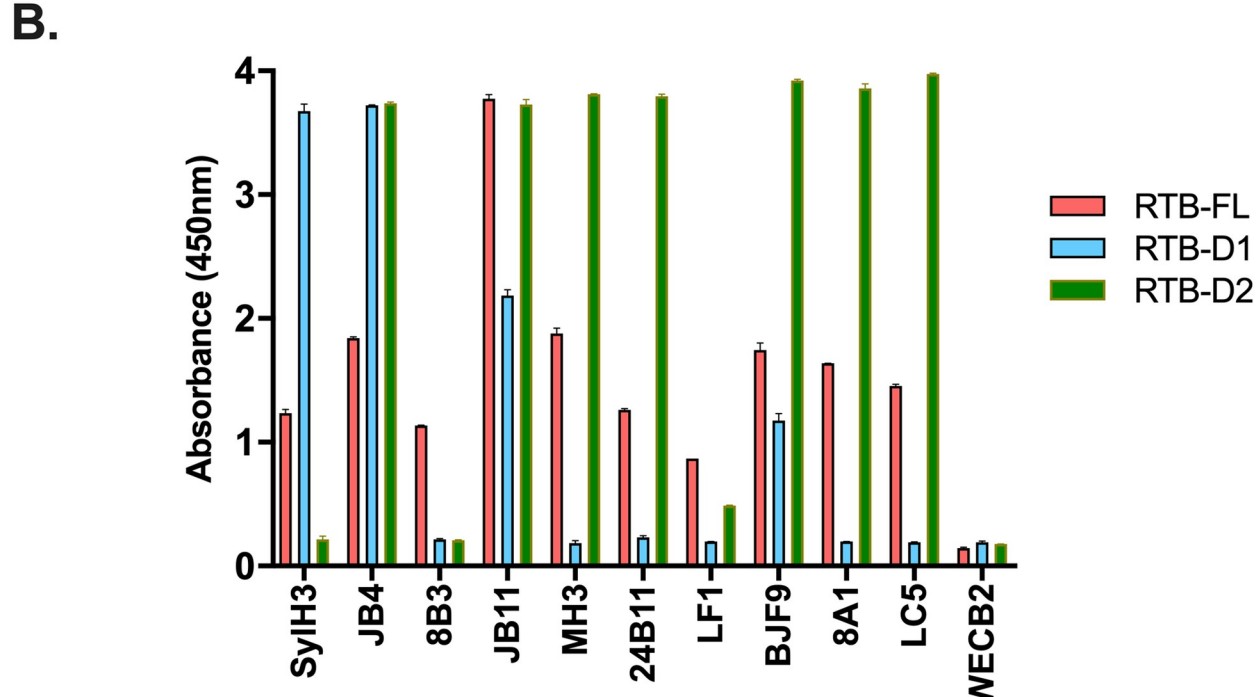

**Fig 2. Epitope localization of RTB-specific mAbs by competition ELISA and RTB domain capture.** (A) RTB-specific mAbs in solution (top; columns) were mixed with biotinylated ricin and then applied to microtiter plates coated with indicated capture mAbs (left; rows). The heatmap

indicates inhibition (%) of ricin capture relative to value obtained in the absence of a competitor. (B) Microtiter plates were coated with indicated mAbs (x axis) and then probed with phage expressing RTB-FL (pink), RTB-D1 (sky blue), or RTB-D2 (olive). The RTA-specific mAb, WECB2, was included as a control.

We found that all 10 RTB-specific mAbs captured M13 phage displaying RTB-FL, albeit with varying degrees of efficiency (**Fig 2B**). The panel of RTB-specific mAbs was then challenged with phage expressing RTB-D1 and RTB-D2 (**Table 1**; **Fig 2B**). As noted above, we have repeatedly predicted that **24B11** recognizes RTB-D1, while **SylH3** recognizes RTB-D2. However, the opposite result was observed. SylH3 captured RTB-D1 in a dose-dependent manner, but did not capture RTB-D2 (**Table 1**; **Fig 2B**; **S1 Fig**). 24B11, on the other hand, captured RTB-D2 in a dose-dependent manner, but did not capture RTB-D1. These unexpected results were not due to technical errors, as the identity of the RTB-D1 and RTB-D2 fusion proteins were confirmed through DNA sequencing. Moreover, the capture ELISA was repeated three times with rederived phage stocks and yielded identical results. Finally, to further confirm specificity of binding, we performed competition ELISAs in which plate bound mAbs (**SylH3** and **24B11**) were treated with saturating amounts of ricin, RTA or RTB before being challenged with RTB-D1 or RTB-D2 phage preparations. The capture of RTB-D1 by SylH3 was inhibited by ricin and RTB but not RTA (**S1 Fig**). Moreover, the capture of RTB-D2 by 24B11 was inhibited by ricin and RTB. We therefore were compelled to reassign SylH3's epitope to RTB-D1 and 24B11's epitope to RTB-D2.

The remaining 8 RTB-specific mAbs aligned as expected relative to SylH3 and 24B11 when tested for the ability to capture phage expressing RTB-D1 and RTB-D2 (**Table 1**; **Fig 2B**). JB4 captured RTB-D1, while the five mAbs that compete with 24B11 by ELISA captured RTB-D2 (MH3, 8A1, JB11, BJF9, LF1) [24, 26, 27]. Thus, the domain assignments for these six mAbs that compete with either SylH3 or 24B11 were internally consistent (**Fig 2**). Based on these results, epitope cluster 5 (SylH3, JB4) was localized to RTB-D1, while epitope cluster 6 (24B11, MH3, 8A1, JB11, BJF9, LF1) was assigned to RTB-D2.

The two remaining mAbs, 8B3 (cluster 5/6) and LC5 (cluster 7), had unusual RTB capture profiles (**Fig 2B**). 8B3 captured RTB-FL but neither of the individual domains (RTB-D1, RTB-D2), suggesting it recognizes an epitope spanning the domain interface. Consistent with that model is the observation that 8B3 competes with both SylH3 and 24B11 (**Table 1**; **Fig 2A**). LC5, on the other hand, captured RTB-D2 but not RTB-D1, even though it did not compete with 24B11. These results indicate that LC5 likely recognizes an epitope on RTB-D2 that is spatially distinct from 24B11. Thus, we assigned epitope cluster 7 to RTB-D2.

It bears noting that JB4, JB11 and BJF9 were each capable of capturing RTB-D1 and RTB-D2 independently (**Table 1**; **Fig 2B**). In the case of JB11, this finding was not unexpected, because there is evidence that it primarily recognizes a linear epitope on RTB-D2 and a second degenerate epitope on RTB-D1[27]. JB4 and BJF9, however, are more perplexing. BJF9 captures RTB-D2 more effectively than RTB-D1, possibly indicating that its epitope spans the two domains with greater contact on domain 2. On the other hand, JB4 captures D1 and D2 equally well even though it competes with SylH3, but not 24B11 or any other D2 mAb. We can only speculate that JB4 recognizes a second, possibly degenerate epitope on D2 and/or its primary epitope spans the D1/D2 interface. Further studies, including co-crystallization trials with ricin toxin, are underway to determine JB4's epitope with more certainty.

## Epitope refinement by competition with RTB-specific V$_H$Hs

To further refine and validate the B cell epitope map of RTB, we performed cross-competition studies with a panel of 12 RTB-specific V$_H$Hs whose epitopes have been tentatively localized

**Table 2.  V$_H$H domain assignments and competition profiles.**

| V$_H$H[a] | Cluster | RTB capture[b] | | | Competition | | | Domain Assignment[c] |
|---|---|---|---|---|---|---|---|---|
| | | FL | D1 | D2 | SylH3 | 24B11 | SyH7 | |
| V5D5 | 5 | - | + | - | +++ | - | - | D1 |
| V5B6 | 5/6 | - | + | - | - | - | - | D1 |
| V5H6 | 5/6 | - | - | - | +++ | - | - | D1-D2 |
| V5E4 | 6 | + | - | +++ | - | - | ++ | D2 |
| V5G1 | 6 | + | - | +++ | - | - | +++ | D2 |
| V5H2 | 6 | + | - | +++ | - | - | +++ | D2 |
| V2C11 | 6 | + | - | +++ | - | - | +++ | D2 |
| V2D4 | 6 | - | - | ++ | - | - | +++ | D2 |
| V4A1 | 6 | - | - | ++ | - | - | +++ | D2 |
| JIZ-B7 | 6/7 | - | - | ++ | - | - | +++ | D2 |
| V5B1 | 6/7 | - | + | - | - | ++ | - | D1 |
| V5C4 | 6/7 | + | - | - | - | ++ | - | D1-D2 |
| V5C1 | 2 | - | - | - | - | - | +++ | RTA |

[a], underlines indicate V$_H$Hs shown to protect Vero cells from ricin toxin, as reported previously [33].

[b],The plus signs (+, ++, +++, etc) summarize the relative amount of RTB capture (RTB-FL; RTB-D1, RTB-D2) from results presented in Fig 3.

[c], "D1-D2" indicates proposed epitope at the D1-D2 interface. Abbreviations: FL, full-length.

on RTB through bootstrapping (e.g., competition ELISAs, RCA-I reactivity) [33, 36] and in some instances X-ray crystallography (M. Rudolph, AY Poon, D. Vance and N. Mantis, *manuscript submitted*). For the cross-competition studies, the 12 V$_H$Hs (four with toxin-neutralizing activity) were competed with each of the 10 RTB-specific mAbs. We also included the RTA-specific mAbs SyH7 and JD4 in the panel. SyH7 recognizes a toxin-neutralizing hotspot known as supercluster 2 (SC2) at the interface between RTA and RTB-D2 [36], while JD4 recognizes an adjacent region of RTA that abuts the RTB-D1/D2 interface [31]. Two V$_H$Hs (V5H6, V5D5) competed with SylH3, seven V$_H$Hs (V5E4, V5G1, V5H2, V2C11, V2D4, V4A1, JIZ-B7) competed with SyH7 and two V$_H$Hs (V5B1, V5C4) competed with 24B11. A final V$_H$H (V5B6) did not compete with any of the three mAbs (**Table 2**; **Fig 3A**).

In an effort to further validate these assignments, the V$_H$Hs were assessed for the ability to capture phage-displayed RTB-D1 and RTB-D2. Indeed, as predicted, the seven V$_H$Hs (V5E4, V5G1, V5H2, V2C11, V2D4, V4A1, JIZ-B7) that competed with SyH7 captured RTB-D2 (**Table 2**; **Fig 3B**). V5D5 captured RTB-D1, a result consistent with its competition with SylH3. The RTB capture assays were less conclusive for the remaining four V$_H$Hs, V5B6, V5H6, V5B1, and V5C4. Nonetheless, the results are consistent with there being three spatially distinct neutralizing hotspots on RTB: one on RTB-D1 defined by SylH3 (cluster 5) and two on RTB-D2, defined by 24B11 (cluster 6) and the RTA-specific mAb, SyH7 (SC2).

### A revised B cell epitope map of RTB

With the information afforded by the RTB-D1/2 capture ELISAs, antibody competition assays and previously reported differential antibody reactivity with RCA-I, we revised our previous B cell epitope map of RTB [24]. For starters, the previous assignment of 24B11's epitope to RTB-D1 and SylH3's epitope to RTB-D2 is incorrect [21, 24, 26, 27]. We can now confidently assign SylH3's epitope to RTB-D1 and 24B11's epitope to RTB-D2 (**Table 1**; **Fig 4**; **S1 Text**).

Within RTB-D1, SylH3's epitope can be further positioned within subdomain 1β-1γ (residues 65–105), based on competition ELISAs and differential reactivity with RCA-I (**S1 Text**;

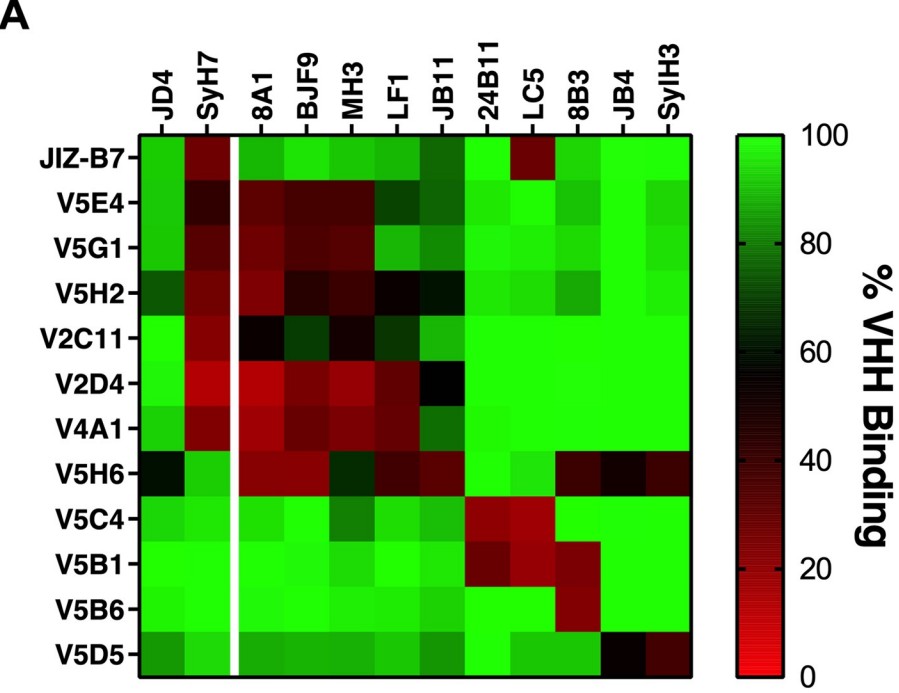

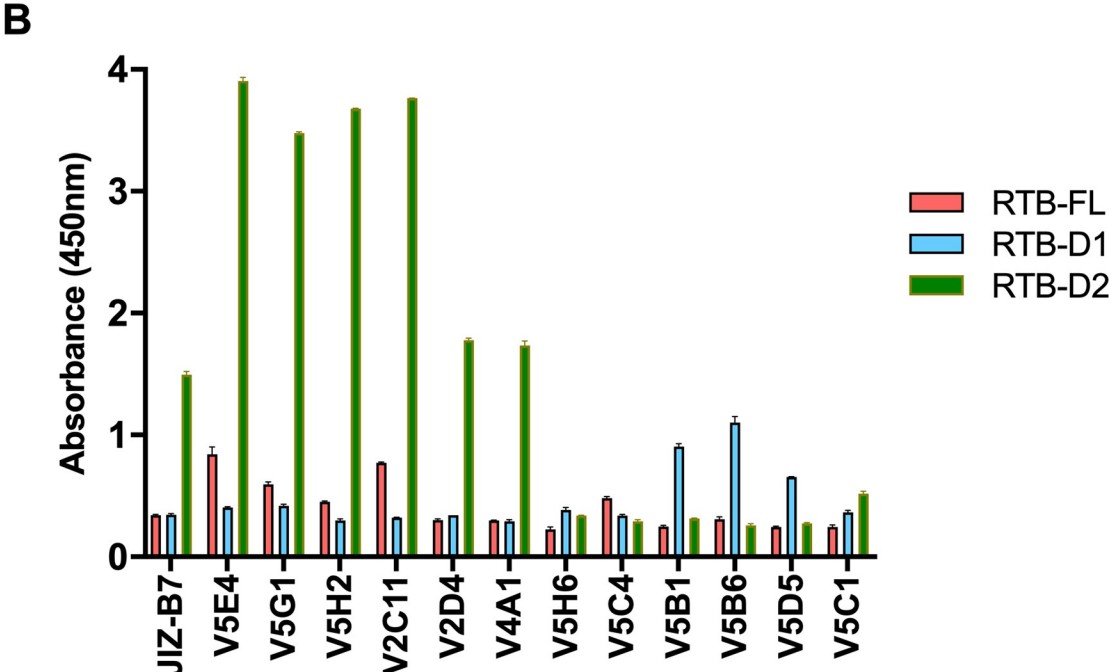

**Fig 3. Epitope localization by competition with RTB-specific V$_H$Hs.** (A) Microtiter plates were coated with RTB-specific mAbs (top; columns) and then saturated with ricin holotoxin before being probed with indicated V$_H$Hs (left; rows). Bound V$_H$Hs were detected with an anti-E-tag-HRP secondary antibody, as described in the Methods. (B) Plate-bound V$_H$Hs were probed with M13 phage expressing RTB-FL (pink), RTB-D1 (sky blue), or RTB-D2 (olive).

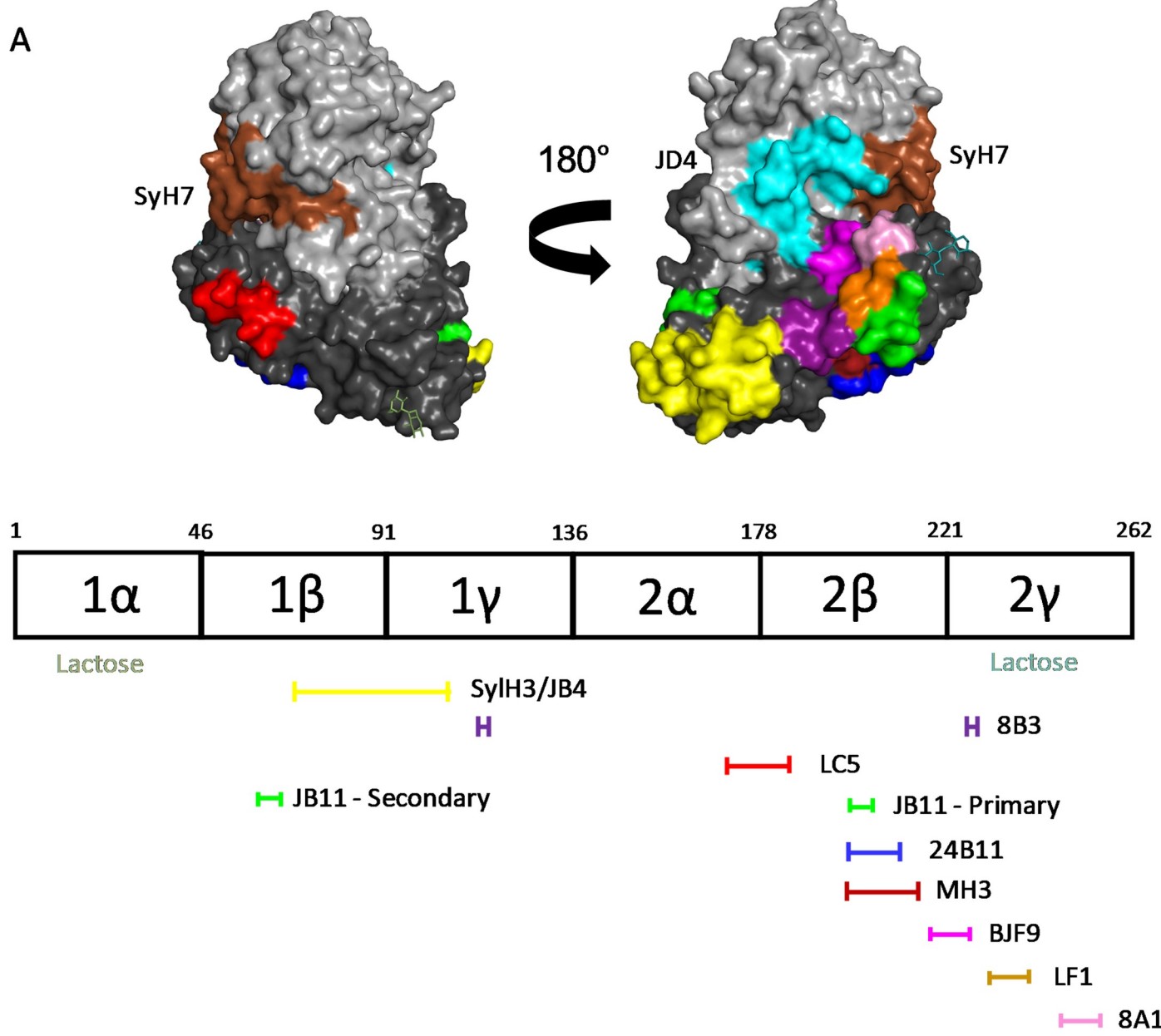

**Fig 4. A Revised B cell epitope map of RTB.** (Top) Surface representation of ricin holotoxin (PDB ID 2AAI) with color coded patches corresponding to proposed location of mAb epitopes. The epitopes recognized by SyH7 and JD4 on RTA were mapped in a previous study by hydrogen exchange mass spectrometry [31]. (Bottom) Linear depiction of RTB's subdomains with horizontal lines below indicating proposed mAb epitope locations. The mAbs are color coded with their respective epitopes shown on the PyMol image above. RTB's two CRDs are located in subdomains 1α and 2γ and highlighted ("lactose").

[24, 27]. By all accounts, JB4's epitope is indistinguishable from SylH3's epitope, even though the two mAbs have different VH and VL CDR1-3 sequences [37]. We conclude therefore that SylH3 and JB4's epitopes collectively define a toxin-neutralizing hotspot on RTB-D1. This region of RTB-D1 is being further interrogated with a collection of recently identified toxin-neutralizing and non-neutralizing $V_H$Hs (A. Poon, D. Vance, and N. Mantis, unpublished results). It remains to be determined whether there are additional targets of vulnerability on RTB-D1.

Within RTB-D2, 24B11's epitope is proposed to reside within subdomain 2β. This conclusion is based on RTB-D1/D2 phage display and competition ELISAs (**S1 Text**). As shown in **Fig 4**, subdomain 2β (residues 178–221) is situated on the underside of RTB and distant from ricin's second Gal/GalNAc pocket in subdomain 2γ. While there are other RTB-specific mAbs in our collection that compete with 24B11, including two (MH3, 8A1) with $IC_{50}$ values similar to 24B11 ($<$5 nM), two (8B3, LF1) with substantially less potent toxin-neutralizing activity ($\sim$30 nM $IC_{50}$) and two (JB11, BJF9) with no notable activity, the $V_HH$ competition profiles shown in **Fig 3** demonstrate that 24B11's epitope is distinct. In fact, the $V_HH$ competition profiles with mAbs MH3, 8A1, BJF9, LF1 and JB11 suggest their epitopes are juxtaposed and possibly overlapping with subdomain 2γ (**Fig 4**). In total, our results suggest that within RTB-D2 there is either a toxin-neutralizing "belt" stretching from subdomains 2β-2γ or two hotspots separated by a trough.

## Discussion

The structurally duplicative and functionally redundant nature of RTB has made B cell epitope identification difficult. In the current study, our ability to successfully express RTB's two individual domains (RTB-D1, RTB-D2) as fusion proteins on the surface of M13 enabled us to localize with confidence SylH3's epitope to RTB-D1 and 24B11's epitope to RTB-D2. These domain assignments are further buttressed by competition assays with mAbs and $V_HHs$ whose epitopes on ricin holotoxin have been resolved by HX-MS and/or X-ray crystallography (M. Rudolph, AY Poon, D. Vance and N. Mantis, *manuscript submitted*) [31, 33]

With the former epitope assignments of SylH3 and 24B11 upended, it is necessary to revisit previous interpretations about mechanisms of toxin-neutralization and the Type I and Type II labels. For example, based on SylH3's ability to inhibit ricin attachment to a variety of primary cell types, including murine alveolar macrophages, bone marrow-derived macrophages, and Kupffer cells, we classified it as Type I and reasoned that it must target an epitope in proximity to the high affinity CRD in RTB subdomain 2γ [26, 38, 39]. However, the results presented herein indicate that SylH3's epitope is situated on RTB-D1, and, specifically within subdomain 1β. This supposition is substantiated by an X-ray crystal structure of SylH3 Fab fragments in complex with ricin holotoxin (M. Rudolph, D. Vance and N. Mantis, *manuscript in preparation*). Unfortunately, by positioning SylH3's epitope outside RTB's two galactoside-specific CRD elements, it is difficult to envision how SylH3 interferes with ricin attachment without invoking a possible role for allostery [40]. Even if SylH3 were to occlude access to CRD 1α, there are numerous studies in the literature demonstrating that 2γ alone is sufficient to promote ricin entry into cells via lactose-dependent and -independent pathways [12–14]. The suggestion that there is a cryptic CRD in subdomain 1β is not entirely dismissible [41], but, at best, it contributes only a small fraction to cell attachment and even fully obstructing this element would not account for SylH3's toxin-neutralizing activity. Irrespective of the mechanism of SylH3 action, the mAb has proven to have value prophylactically and therapeutically alone and when combined with an RTA-specific mAb, PB10, and used in mouse models of pulmonary ricin intoxication [39, 42].

24B11 is representative of the type II class of RTB-specific mAbs, defined by robust toxin-neutralizing activity that is not attributable to inhibition of toxin attachment. Rather, 24B11 IgG and Fab fragments, when bound to ricin on the cell surface, interrupt toxin trafficking from the plasma membrane to the TGN by shunting the complex for lysosomal degradation [25]. Based on linear epitope mapping studies, we have been working under the assumption that 24B11 recognizes an epitope in proximity to CRD 1α [21]. However, reassignment of 24B11's epitope to RTB-D2 is actually more consistent its functional profile. Specifically,

mAbs (e.g., SyH7) and V$_H$Hs (e.g., V5E4, V2C11) that bind at the interface of RTA and RTB's subdomain 2γ affect ricin retrograde trafficking in Vero and HeLa cells (M. Rudolph, A. Poon, D. Vance, N. Mantis, *manuscript submitted*) [43]. Our results position 24B11's epitope just outside of SC2, which is defined operatively as competition with SyH7 [36]. Collectively, these results suggest that RTB subdomain 2γ and neighboring elements are involved in sorting ricin within early endosomes, possibly by interacting with one or more host factors [44]. In conclusion, we have localized sites of vulnerability on RTB-D1 that appear to be primarily involved in toxin attachment to host cells, and on RTB-D2 (and spilling over onto RTA) that apparently function in intracellular trafficking.

## Supporting information

**S1 Fig. RTB-D1/D2 capture and competition ELISAs with SylH3 and 24B11.** (A) Dose-dependent capture of RTB-D1 and RTB-D2 by plate-bound SylH3 and 24B11. As described in the Material and Methods, microtiter plates were coated with indicated mAbs (1 μg/mL in PBS) then probed with indicated number of RTB-D1 or RTB-D2 plaque forming units (PFU) per mL. The plates were probed with anti-M13-HRP secondary antibody to detect bound phage. Shown is a single ELISA with three technical replicates; (B) Specificity of RTB-D1 or RTB-D2 capture ELISAs. Microtiter plates coated with indicted mAb were blocked with 2% (w/v) bovine serum albumin (BSA) in PBS then incubated with 10 μg/mL of ricin, RTA or RTB before being probed with 5 x 10$^{10}$ PFU per well of RTB-D1 or RTB-D2. The plates were developed with anti-M13-HRP secondary antibody and TMB, as noted above.
(JPG)

**S1 Table. PCR primers used in this study.**
(PDF)

**S2 Table. Primers used for amplification of RTB domain.**
(PDF)

**S1 Text. Details associated with epitope assignments on RTB.**
(PDF)

## Acknowledgments

We gratefully acknowledge the Wadsworth Center's Media and Tissue Culture core facility for bacterial media and the Applied Genomic Technologies Core for DNA sequencing services. We thank Elizabeth Cavosie (Wadsworth Center) for administrative assistance and Dr. Michael Rudolph (New York Structural Biology Center) for helpful discussions and for sharing preliminary findings.

## Author Contributions

**Conceptualization:** David J. Vance, Nicholas J. Mantis.

**Formal analysis:** Nicholas J. Mantis.

**Funding acquisition:** Nicholas J. Mantis.

**Investigation:** David J. Vance, Amanda Y. Poon, Nicholas J. Mantis.

**Methodology:** David J. Vance.

**Project administration:** David J. Vance, Nicholas J. Mantis.

**Supervision:** Nicholas J. Mantis.

**Writing – original draft:** David J. Vance, Amanda Y. Poon, Nicholas J. Mantis.

**Writing – review & editing:** David J. Vance, Amanda Y. Poon, Nicholas J. Mantis.

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
