## [Decision Letter · Decision Letter 0]

12 Aug 2020

PONE-D-20-21014

Sites of Vulnerability on Ricin B Chain Revealed through Epitope Mapping of Toxin-Neutralizing Monoclonal Antibodies

PLOS ONE

Dear Dr. Mantis,

Thank you for submitting your manuscript to PLOS ONE. After careful consideration, we feel that it has merit but does not fully meet PLOS ONE’s publication criteria as it currently stands. Therefore, we invite you to submit a revised version of the manuscript that addresses the points raised during the review process.

We look forward to receiving your revised manuscript.

Kind regards,

Kevin A. Henry

Academic Editor

PLOS ONE

Additional Editor Comments:

Please pay special attention to comments from reviewer #2, who felt these changes would be major rather than minor.

Journal Requirements:

Reviewers' comments:

Reviewer's Responses to Questions

**Comments to the Author**

1. Is the manuscript technically sound, and do the data support the conclusions?

Reviewer #1: Yes

Reviewer #2: Partly

Reviewer #3: Yes

2. Has the statistical analysis been performed appropriately and rigorously? 

Reviewer #1: N/A

Reviewer #2: N/A

Reviewer #3: Yes

3. Have the authors made all data underlying the findings in their manuscript fully available?

Reviewer #1: Yes

Reviewer #2: Yes

Reviewer #3: Yes

4. Is the manuscript presented in an intelligible fashion and written in standard English?

Reviewer #1: Yes

Reviewer #2: Yes

Reviewer #3: Yes

5. Review Comments to the Author

Reviewer #1: I have reviewed the manuscript entitled “Sites of vulnerability on Ricin B chain revealed through epitope mapping of toxin-neutralizing monoclonal antibodies” (PON-E-20-21014). In this manuscript the authors have employed recombinant protein and subdomains to identify the epitopes engaged through type I and II mAbs. Following their identity additional mAbs (neutralizing and non-neutralizing) and VHHs were used to compete the epitopes in competitive binding assays. These results help delineate the domain specificity and are suggestive of their independent contributions to distinct steps in the intoxication pathway through this structural study. Following this review the recommendation is accept without revisions.

Reviewer #2: Ricin toxin belongs to the A-B toxin family, with the ricin B subunit (RTB) responsible for cell attachment and intracellular trafficking. RTB has attracted considerable interest as a target for neutralizing antibodies, however most of the epitopes recognized by neutralizing and non-neutralizing epitopes have been only roughly mapped. In this work, the authors localize the epitopes of two neutralizing mAbs, 24B11 and Sy1H3 within the RTB. These previously characterized antibodies represent two mechanistically distinct classes of neutralizing mAbs against Ricin Toxin B: SyH3-like mAbs (type I) that prevent cells attachment and 24B11-like mAbs (type II) that prevent intracellular transport of the toxin. The authors individually expressed the related domains RTB-D1 and RTB-D2 separately as M13 phage pIII fusion proteins and performed phage ELISAs to assign Sy1H3 binding to RTB-D1 and 24B11 to RTB-D2. The binding domain of other RTB neutralizing mAbs was similarly assigned and their competition profiles relative to Sy1H3 and 24B11 were also determined. To further refine the epitopes of Sy1H3 and 24B11, the authors performed competition ELISAs with a panel VHHs whose epitopes are better described in previous work. Overall, this study proposes to re-define the epitopes of two well characterized and mechanistically distinct neutralizing mAbs further informing the location of Ricin B protective epitopes. While important, these conclusions contradict prior work yet rely upon a single experimental approach using phage-displayed proteins which were not characterized for proper folding. If the authors propose changing previously described epitope maps, it is important to include two complementary experimental approaches with proper control data that support the same new conclusion.

Major comments

1. The data would be more compelling if at least some of the data are shown with the antibody dilution curve to allow readers to evaluate the dynamic range of the assays and visualize the degree of competition used to derive the +, ++ and +++ metrics shown in the tables. I am surprised that the phage ELISAs were performed with one phage concentration, 5x1010 pfu per well and did not serially dilute the phage to (a) visualize the dose-response and (b) use these data to rank clones by relative affinity as a secondary check on data consistency with other methods.

2. I am perplexed by the statement on lines 100-103 that there is a “proven inability to express recombinant RTB in E. coli” however, “full-length RTB as well as RTB-D1 and D2 constructs have been successfully expressed as fusion proteins on the tip of filamentous phage M13.” These statements seem contradictory since M13 phage fusion proteins are also produce recombinantly in E. coli and exported from the cytoplasm to the periplasm by Sec machinery before phage assembly. What tests have been performed to assess the structural and functional integrity of the phage displayed RTB and how does recombinantly expressed soluble RTB perform on these same tests? Recombinant RTB produced in E. coli is commercially available, which suggests (but does not guarantee) that it is functional. The rational for using phage displayed RTB should use more robust logic and controls to determine whether (or the extent to which) the displayed proteins are properly folded and active shown.

Minor comments

1. Line 156: single domain Vhh ELISas were performed

2. Add a sentence of the VHH construct (i.e. includes E-tag which is then used for detection in ELISAS)

3. Tables 1 and 2 define FL (Full length)

Reviewer #3: This paper describes epitope mapping for two neutralizing mAbs targeting Ricin's B-subunit. Mapping the epitopes for this region is hard because the two B subunits, RTB-D1 and RTB-D2, are highly homologous.

I would describe this paper as technically sound and well presented, which makes it suitable for publication in PLoS as-is in my view. My only 'criticism' would be that it is highly incremental relative to the author's previous work in the area. Having said that, the paper gets results from an analytically challenging system and corrects an earlier (less confident) report by the same authors that misidentified the epitope for one of the two antibodies, and for this, the authors should be commended.

6. PLOS authors have the option to publish the peer review history of their article (what does this mean?). If published, this will include your full peer review and any attached files.

Reviewer #1: No

Reviewer #2: No

Reviewer #3: **Yes: **Derek J. Wilson

---

## [Author Response · Author response to Decision Letter 0]

4 Oct 2020

Reviewer #1: Following this review the recommendation is accept without revisions.

Response: We thank Reviewer 1 for his/her favorable review of the manuscript.

Reviewer #2: Overall, this study proposes to re-define the epitopes of two well characterized and mechanistically distinct neutralizing mAbs further informing the location of Ricin B protective epitopes. While important, these conclusions contradict prior work yet rely upon a single experimental approach using phage-displayed proteins which were not characterized for proper folding. If the authors propose changing previously described epitope maps, it is important to include two complementary experimental approaches with proper control data that support the same new conclusion.

Response: The Reviewer makes a valid point in noting the importance of a secondary methodology to confirm/validate the proposed epitope locations describe in this manuscript. The most compelling evidence in this respect are the X-ray co-crystal structures of SylH3 and V5E4 in complex with ricin holotoxin. The two recently refined structures confirm SylH3 and V5E4’s proposed epitope location, as shown in Figure 4. There are two separate manuscripts nearing submission that will describe these structures. To use these structures as a means to validate the current study, we will link the current manuscript to those future manuscripts using “Crossref,” which is offered as part of PLOS publications. Of course, the structures will also be available on the PDB.

1. The data would be more compelling if at least some of the data are shown with the antibody dilution curve to allow readers to evaluate the dynamic range of the assays and visualize the degree of competition used to derive the +, ++ and +++ metrics shown in the tables. I am surprised that the phage ELISAs were performed with one phage concentration, 5x10^10 pfu per well and did not serially dilute the phage to (a) visualize the dose-response and (b) use these data to rank clones by relative affinity as a secondary check on data consistency with other methods.

Response: As requested by the Reviewer, we have now included S1 Figure showing a dose-response curve of SylH3 and 24B11 capturing RTB-D1 and RTB-D2, respectively. The data demonstrate that 5x10^10 PFU is within the dynamic range of the dose-response curve. Moreover, also included in that figure are competition assays with ricin, RTB and RTA to demonstrate specificity of the RTB-D1 and RTB-D2 phage. 

2. I am perplexed by the statement on lines 100-103 that there is a “proven inability to express recombinant RTB in E. coli” however, “full-length RTB as well as RTB-D1 and D2 constructs have been successfully expressed as fusion proteins on the tip of filamentous phage M13.” These statements seem contradictory since M13 phage fusion proteins are also produce recombinantly in E. coli and exported from the cytoplasm to the periplasm by Sec machinery before phage assembly. What tests have been performed to assess the structural and functional integrity of the phage displayed RTB and how does recombinantly expressed soluble RTB perform on these same tests? Recombinant RTB produced in E. coli is commercially available, which suggests (but does not guarantee) that it is functional. The rational for using phage displayed RTB should use more robust logic and controls to determine whether (or the extent to which) the displayed proteins are properly folded and active shown.

Response: While the Reviewer is correct in that RTB has been expressed in E.coli when targeted to the periplasmic space, the overall yield of properly folded protein is low and highly strain dependent {Hussain, 1989; Wales, 1994}. Moreover, mutagenesis of RTB has only been done in eukaryotic expression systems such as Xenopus oocytes (Newton, 1992). However, as reported by Swimmer (1982), full length and individual domains of RTB are amenable to M13 display following expression (and secretion) by E.coli. Considering our lab’s expertise in M13-phage display, we chose to that as the basis for the current study.

To address the very astute comment regarding “…. tests have been performed to assess the structural and functional integrity of the phage displayed RTB and how does recombinantly expressed soluble RTB perform on these same tests?” we now provide competition assays with native ricin and RTB (S1 Figure).

Minor comments

1. Line 156: single domain Vhh ELISas were performed. Response: Corrected

2. Add a sentence of the VHH construct (i.e. includes E-tag which is then used for detection in ELISAS). Response: As requested, we have modified the text to state “The VHHs used in this study carry a C-terminus E epitope tag (E-tag; GAPVPYPDPLEPR) for the purpose of detection by ELISA using HRP-conjugated, affinity-purified anti-E-tag rabbit IgG.

3. Tables 1 and 2 define FL (Full length). Response: Corrected

Reviewer #3: My only 'criticism' would be that it is highly incremental relative to the author's previous work in the area. Having said that, the paper gets results from an analytically challenging system and corrects an earlier (less confident) report by the same authors that misidentified the epitope for one of the two antibodies, and for this, the authors should be commended.

Response: We thank Reviewer 3 for his/her favorable review of the manuscript.

---

## [Decision Letter · Decision Letter 1]

27 Oct 2020

Sites of Vulnerability on Ricin B Chain Revealed through Epitope Mapping of Toxin-Neutralizing Monoclonal Antibodies

PONE-D-20-21014R1

Dear Dr. Mantis,

We’re pleased to inform you that your manuscript has been judged scientifically suitable for publication and will be formally accepted for publication once it meets all outstanding technical requirements.

Kind regards,

Kevin A. Henry

Academic Editor

PLOS ONE

Additional Editor Comments (optional):

Reviewers' comments:

Reviewer's Responses to Questions

**Comments to the Author**

1. If the authors have adequately addressed your comments raised in a previous round of review and you feel that this manuscript is now acceptable for publication, you may indicate that here to bypass the “Comments to the Author” section, enter your conflict of interest statement in the “Confidential to Editor” section, and submit your "Accept" recommendation.

Reviewer #2: All comments have been addressed

2. Is the manuscript technically sound, and do the data support the conclusions?

Reviewer #2: Yes

3. Has the statistical analysis been performed appropriately and rigorously? 

Reviewer #2: Yes

4. Have the authors made all data underlying the findings in their manuscript fully available?

Reviewer #2: Yes

5. Is the manuscript presented in an intelligible fashion and written in standard English?

Reviewer #2: Yes

6. Review Comments to the Author

Reviewer #2: Thank for you for the additional methods and Supp Fig 1 which I think will help this very well written paper be more useful to other scientists.

7. PLOS authors have the option to publish the peer review history of their article (what does this mean?). If published, this will include your full peer review and any attached files.

Reviewer #2: No

---

## [Editor Report · Acceptance letter]

30 Oct 2020

PONE-D-20-21014R1 

Sites of Vulnerability on Ricin B Chain Revealed through Epitope Mapping of Toxin-Neutralizing Monoclonal Antibodies 

Dear Dr. Mantis:

I'm pleased to inform you that your manuscript has been deemed suitable for publication in PLOS ONE. Congratulations! Your manuscript is now with our production department. 

Kind regards, 

on behalf of

Dr. Kevin A. Henry 

Academic Editor

PLOS ONE